# Dynamic ICSP Graph Optimization Approach for Car-Like Robot Localization in Outdoor Environments †



**Zhan Wang** [1,*] 📷 **, Alain Lambert** [2] 📷 **and Xun Zhang** [1]

[1]  Institut Supérieur D'électronique de Paris, 75006 Paris, France
[2]  Laboratoire de Recherche en Informatique, CNRS, Univ Paris-Sud, Université Paris-Saclay, 91400 Orsay, France
*  Correspondence: zhan.wang@u-psud.fr
†  This paper is an extended version of our paper published in Proceedings of the 10th Computer Science and Electronic Engineering Conference (CEEC), Colchester, UK, 19–21 September 2018.

**Abstract:** Localization has been regarded as one of the most fundamental problems to enable a mobile robot with autonomous capabilities. Probabilistic techniques such as Kalman or Particle filtering have long been used to solve robotic localization and mapping problem. Despite their good performance in practical applications, they could suffer inconsistency problems. This paper presents an Interval Constraint Satisfaction Problem (ICSP) graph based methodology for consistent car-like robot localization in outdoor environments. The localization problem is cast into a two-stage framework: visual teach and repeat. During a teaching phase, the interval map is built when a robot navigates around the environment with GPS-support. The map is then used for real-time ego-localization as the robot repeats the path autonomously. By dynamically solving the ICSP graph via Interval Constraint Propagation (ICP) techniques, a consistent and improved localization result is obtained. Both numerical simulation results and real data set experiments are presented, showing the soundness of the proposed method in achieving consistent localization.

**Keywords:** localization; mapping; ICSP graph; interval constraint satisfaction problem; interval constraint propagation

## 1. Introduction

Localization is a problem for mobile robots to localize themselves in the environment with sensory information from their embedded sensors. A reliable solution to this problem is the prerequisite and foundation for achieving high level applications, such as tracking, path planning, security alerts, environmental survey, etc. A vast number of works dedicated to the localization problems utilize the probabilistic method, which is based on the propagation of probabilistic distributions of the sensor noise and the unknown parameters (the robot pose state). While some others get rid of dealing with the uncertainty probability distributions, they assumed that noise is unknown but bounded by real intervals. Those methods provide solutions presented by a set of bounded configurations in which the robot is guaranteed to be, and they are classified as deterministic methods.

Probabilistic techniques have long been used in solving robotic localization and mapping issues. The most commonly used methods are Kalman filtering and Particle filtering [1–3]. However, these methods suffer inconsistency problems in some scenarios. This shortcomings have long been noticed by the research community [4–8]. An interval analysis based method is another type of method. Different from probabilistic methods, which make hypotheses on the probability distribution, these methods take a soft assumption that all the noise is bounded within known limits. This is a realistic representation because many manufacturers provide the maximum and minimum errors for the

sensors they produce. These maximum and minimum error values can be regarded as the error bounds. Based on these error bounds, interval analysis based methods can recursively propagate such bounded errors by using consistency techniques and systematic search methods. Contrary to probabilistic methods, interval analysis based methods calculate guaranteed sets to enclose the real solutions, without losing any feasible values.

Interval analysis based methods have achieved applications successfully such as parameter and state estimation [9,10] as well as the robotic localization and mapping field. Seignez [11] presents their simulation results of a mobile robot navigating in an indoor environment. The ability of an interval analysis method to cope with erroneous data and to obtain consistent estimations of the robot pose was demonstrated. Lambert [12] extended such work for an outdoor vehicle equipped with two proprioceptive sensors and a GPS receiver. Comparison was made with the Particle Filter localization, showing the better performance of an interval based method in terms of consistency issues. Gning [13] and Kueviakoe [14] deal with the outdoor localization problem in the framework of an Interval Constraint Satisfaction Problem (ICSP). Those works used interval constraint propagation techniques to fuse the redundant data of sensors [9]. Drevelle [15] proposed a relaxed constraint propagation approach to deal with erroneous GPS measurements. Bonnifait [16] combined constraint propagation and set inversion techniques and presented a cooperative localization method with significant enhancement in terms of accuracy and confidence domains. Experimental results illustrate that the precision obtained is very good with a consistent localization, and the constraint propagation techniques are well adapted to real time implementation.

The main advantage of interval analysis based localization over Kalman filtering or Bayesian methods is that they provide guaranteed solutions without the need to linearize the robot motion or the sensing models, unlike the probabilistic counterparts that require linearization to facilitate the propagation of uncertainties. Moreover, interval methods do not assume any noise probability distribution in the system; they just require a soft assumption about the support of the noise, i.e., it is bounded by real intervals. Thus, they can provide guaranteed and consistent results.

We present, in this paper, a dynamic ICSP graph optimization based methodology for solving outdoor robotic localization problems. We put forward a two-stage framework: visual teach and repeat. Closer approaches have already been proposed by different researchers [17–19], which all involve a visual learning step to reconstruct a map of the environment, and then use this map for localization and navigation tasks. Royer [17] uses bundle adjustment in the mapping step and the localization results are obtained via a Newton iteration method. Lim [19] uses structure from a motion (SFM) algorithm to reconstruct the 3D environment; then, in the localization step, two discrete Kalman filters are employed to estimate the camera trajectory. Courbon's method [18] doesn't involve the 3D reconstruction procedure during the map building step; instead, it records some key views and the performed path as references and uses them for future navigation missions. Our proposed method uses totally different theory and techniques from these approaches; we use a bounded-error model to parameterize landmarks detected by a monocular camera and cast the landmark estimation process as an ICSP. The ICSP graph architecture is put forward for intensively interval domain contraction. By using consistent techniques (ICP) to search for all possible solutions, a consistent real-time localization result is obtained. This result could be further improved by performing global optimization over the ICSP graph.

The paper is organized as follows: Section 2 introduces the basics of interval analysis and constraint propagation. Section 3 presents the formulation of ICSP based landmark estimation, and details the construction and resolution of the ICSP graph. The real-time localization process is introduced in Section 4. Numerical and experimental results are given in Section 5. Section 6 concludes the paper and proposes the perspective of future work.

## 2. Overview of Interval Analysis

Interval analysis and the constraints satisfaction problem are the two main mathematical theories on which our work are based. In this section, we give a brief description of the fundamental knowledge that are used in our work.

### 2.1. Principle of Interval

Interval analysis [20,21] was developed in the 1960s in order to deal with approximation problems encountered during calculation. It is a numerical method which puts bounds on rounding or measurements errors during mathematical computations in order to get reliable results.

An interval is a connected subset of $\mathbb{R}$, defined as $[x] = [\underline{x}, \bar{x}] = \{x \in \mathbb{R} \mid \underline{x} \leq x \leq \bar{x}\}$, where $\underline{x}$ and $\bar{x}$ are respectively the lower and upper bound of $[x]$, $\underline{x}, \bar{x} \in \mathbb{R}$ and $\underline{x} \leq \bar{x}$. $[-2, 2]$, $[3, +\infty]$, $[5, 5]$ are some interval examples. The empty set $\varnothing$ is also considered as an interval since it can be used to represent the null solution to a problem. The width of $[x]$ is computed by $w([x]) = \bar{x} - \underline{x}$. The set of intervals is denoted by $\mathbb{IR}$.

An interval vector (or box) is a generalization of the interval concept. It is a vector whose components are intervals. $[\mathbf{x}]$ ($[\mathbf{x}] \in \mathbb{IR}^n$) is an $n$-dimensional interval vector defined as the Cartesian product of $n$ intervals:

$$[\mathbf{x}] = [x_1] \times [x_2] \times \cdots \times [x_n], \tag{1}$$

where the $i^{th}$ interval $[x_i]$ ($i = 1, \cdots, n$) is the projection of $[\mathbf{x}]$ on to the $i^{th}$ axis. The volume of the interval vector is then computed via

$$vol([\mathbf{x}]) = \prod_{1 \leq i \leq n} w([x_i]). \tag{2}$$

The volume of the interval box is usually used to evaluate the uncertainty of $[\mathbf{x}]$ [22].

For instance, the configuration of a vehicle's pose usually contains three parameters: position $x$, position $y$ and heading angle $\theta$. Consequently, for vehicle localization, the solution is a three-dimensional box: $[x] \times [y] \times [\theta]$. When projected on the $x - y$ plane, it decreases to a two-dimensional box, which gives the position region of the vehicle. The projection on the $x$ axis and $y$ axis then gives the $x$ position and $y$ position of the vehicle separately—see the illustration on Figure 1.

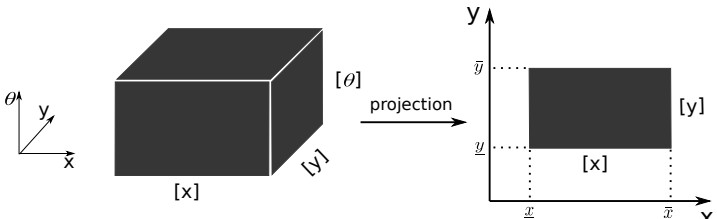

**Figure 1.** Robot localization box.

### 2.2. Operations of Interval Arithmetic

Rules have been defined to apply the basic arithmetical operations on intervals. Consider two intervals $[\mathbf{x}]$ and $[\mathbf{y}]$ and a binary operator $\diamond \in \{+, -, \times, \div\}$, the smallest interval which contains all feasible values for $[\mathbf{x}] \diamond [\mathbf{y}]$ is defined as:

$$[\mathbf{x}] \diamond [\mathbf{y}] := \{x \diamond y \mid x \in [\mathbf{x}], y \in [\mathbf{y}]\}. \tag{3}$$

$\diamond$ is a complete operator as it returns all the possible solutions for the computations between the two intervals. The resulting solutions are thus guaranteed. For example, $[-2, 3] + [1, 2] = [-1, 5]$, $[-2, 4] \times [-2, 2] = [-8, 8]$.

The image of a vector function $\mathbf{f} : \mathbb{IR}^m \to \mathbb{IR}^n$ (defined by arithmetical operators and elementary functions), over an interval box $[\mathbf{x}]$ can be evaluated by its inclusion function $[\mathbf{f}]$, whose output contains all possible values taken by $\mathbf{f}(\cdot)$ over $[\mathbf{x}]$:

$$\forall [\mathbf{x}] \in \mathbb{IR}^m, \mathbf{f}([\mathbf{x}]) \subset [\mathbf{f}]([\mathbf{x}]). \tag{4}$$

The image set of $\mathbf{f}([\mathbf{x}])$ may have any shape; the inclusion function $[\mathbf{f}]$ of $\mathbf{f}$ makes it possible to compute a box that is guaranteed to contain the entire image set (see Figure 2). The inclusion function is one of the most important notions in interval analysis. It can be used to represent equations. Such equations, also called constraints, are the core of the Interval Constraint Satisfaction Problem.

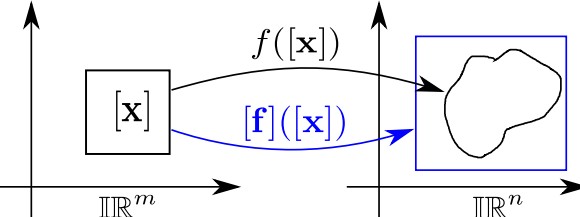

**Figure 2.** Image of an interval box by function $f$ and its inclusion function $[\mathbf{f}]$.

### 2.3. Interval Constraint Satisfaction Problem (ICSP)

The Constraint Satisfaction Problem (CSP) is a mathematical problem of finding a solution to a set of variables whose states should satisfy a number of constraints. The notion of an Interval Constraint Satisfaction Problem (ICSP) was introduced by Hyvoen [23] in 1992. It is a certain form of CSP that deals with continuous variable domains represented by intervals. It is typically defined as a triple ICSP($V, D, C$):

- a set $V$ of $n$ variables $\{v_1, v_2, \ldots, v_n\}$,
- a set $D$ of $n$ domains $\{d_1, d_2, \ldots, d_n\}$, such as for each variable $v_i$, a domain $d_i$ with the possible values for that variable is given. $d_i$ could be an interval or union of intervals;
- a set $C$ of $p$ constraints $\{c_1, c_2, \ldots, c_p\}$, each constraint $c_i$ defines a relationship over a subset variables of $V$, e.g., $c_1(v_1, v_2, v_3) = 0$ restricts the possible domains of $v_1$, $v_2$ and $v_3$.

ICSP is a mathematical problem of searching all the possible domains of the variables that satisfy all the constraints corresponding to the ICSP.

### 2.4. Contractor

The concept of contractor is directly inspired from the ubiquitous concept of filtering algorithms in constraint programming. Given a constraint $c$ relating a set of variables $\mathbf{x}$, an algorithm $\mathcal{C}$ is called a contractor (or a filtering algorithm) if $\mathcal{C}$ returns a sub-domain of the input domain $[\mathbf{x}]$ and the resulting sub-domain $\mathcal{C}([\mathbf{x}])$ contains all the feasible values with respect to the constraint $c$. Mathematically,

$$\forall [\mathbf{x}] \in \mathbb{IR}^n, \mathcal{C}([\mathbf{x}]) \subseteq [\mathbf{x}], \tag{5}$$

$$\forall x \in [\mathbf{x}], c(x) \Rightarrow x \in \mathcal{C}([\mathbf{x}]). \tag{6}$$

The first condition indicates the contractance property and the second one refer to the correctness. The concept has been formalized and intensively used in solving the constraint satisfaction problems. Contractor has been formalized and intensively used in solving ICSP, such as Forward/Backward, HC4, etc.

*2.5. Interval Consistency*

The notion of consistency is the fundamental and underlying concept to the domain contraction for the Interval Constraint Satisfaction Problem.

**Local consistency:** When an algorithm locally computes the domain of variables, considering only one constraint at a time, then we say the algorithm is local consistency (related to the constraint). Since it doesn't take into account the whole constraint system, it may not achieve the optimal solution for all the variables. The Forward/Backward contractor is a classical local consistency algorithm since it calculates the primitive constraints one by one. The strength of local consistency algorithm is their good performance in computing time. Due to the simple calculation (one constraint at a time), the local consistency algorithm always surpasses the global ones in the comparison of computing time.

**Global consistency:** An algorithm is said to have global consistency if it takes into account of all the constraints of the system when performing domain contractions. Consider an *ICSP* with $n$ variables ($V = \{v_1, v_2, \cdots, v_n\}$) and $p$ constraints ($C = \{c_1, c_2, \cdots, c_p\}$). Denote each constraint by $c_j : f_j(V_j) = 0$, where $V_j$ is a subset of $V$ with $j \in \{1, \cdots, p\}$. The domain of $v_i$ is global consistent if the following condition is satisfied:

$$\forall v_i \in [v_i], \ \exists v_1 \in [v1], \cdots, v_{i-1} \in [v_{i-1}], v_{i+1} \in [v_{i+1}], \cdots, v_n \in [v_n], \tag{7}$$

such that $\forall j \in \{1, \cdots, p\}, f_j(V_j) = 0$. If the domain of all variables reaches global consistency, then the contraction algorithm is said to be a global one. A global consistency algorithm seeks the strength of the domain contractions rather than the shortness of computation time.

## 3. Visual Teach Mapping

*3.1. Framework of Visual Teach*

The objective of the visual teach phase is to construct a map of the surrounding environment. The general framework of our proposed visual teach is described in Figure 3. It includes two main processes: motion estimation and observation processing. The two processes are conducted at each timestep. In the motion estimation process, the preliminary pose of the robot is estimated by fusing the measurements of odometric sensors and GNSS receivers; then, in the observation process, the measurement (new captured image) from the camera is processed. By constructing and solving the ICSPs successively, the position of each landmark can be estimated.

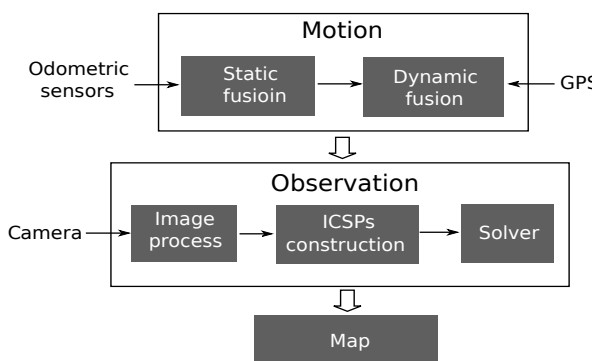

**Figure 3.** Visual teach framework.

*3.2. Motion Estimation*

The motion estimation process integrates the odometric measurements and GNSS data to determine the current robot pose $\mathbf{x}_k$ given the previous pose $\mathbf{x}_{k-1}$ and the sensor data collected during the instant $k-1$ and $k$. It is a dynamic nonlinear state estimation problem. Based on the

definition of ICSP, we can cast the motion estimation process as an ICSP. At timestep $k$, the new $\text{ICSP}_k(V_k, D_k, C_k)$ is generated and defined as

- a set of variables, $V_k$:

$$V_k = \{x_k, y_k, \theta_k, x_{k-1}, y_{k-1}, \theta_{k-1}, \delta s_k, \delta \theta_k\}, \tag{8}$$

- a set of domains, $D_k$:

$$D_k = \{[x_k], [y_k], [\theta_k], [x_{k-1}], [y_{k-1}], [\theta_{k-1}], [\delta s_k][\delta \theta_k]\}, \tag{9}$$

- a set of constraints, $C_k$:

$$C_k = (C_k^1, C_k^2, C_k^3)^T, \tag{10}$$

$$\begin{cases} C_k^1 \Leftrightarrow x_k & = & x_{k-1} + \delta s_k \cos(\theta_{k-1} + \frac{\delta \theta_k}{2}) \\ C_k^2 \Leftrightarrow y_k & = & y_{k-1} + \delta s_k \sin(\theta_{k-1} + \frac{\delta \theta_k}{2}) \\ C_k^3 \Leftrightarrow \theta_k & = & \theta_{k-1} + \delta \theta_k. \end{cases} \tag{11}$$

$C_k$ represents the set of constraints; it is the robotic kinematic model defined by Seignez [11].

Once the ICSP is generated, the robot pose $\mathbf{x}_k$ can be estimated by applying the ICP algorithms over the constraints deduced from the kinematic model. Table 1 gives the equations that are obtained from a Forward/Backward propagation. Interval analysis is a pessimistic method maintaining all possible solutions; the cumulative odometric errors will result in interval expansion, i.e., the width (or uncertainty) of $[\mathcal{X}_k]$ will become larger and larger. This would be a hindrance to pursuing a precise map. To deal with such problem, GPS measurement $[\mathcal{G}_k] = ([g_x], [g_y])^T$ is used to initialize the priori domains of $[\mathcal{X}_k]$.

**Table 1.** Forward/Backward propagation.

| Forward Propagation | | |
| --- | --- | --- |
| 1. $[x_k]$ | $=$ | $[x_k] \bigcap ([x_{k-1}] + [\delta s_k] \cos([\theta_{k-1}] + \frac{[\delta \theta_k]}{2}))$ |
| 2. $[z_k]$ | $=$ | $[z_k] \bigcap ([z_{k-1}] + [\delta s_k] \sin([\theta_{k-1}] + \frac{[\delta \theta_k]}{2}))$ |
| 3. $[\theta_k]$ | $=$ | $[\theta_k] \bigcap ([\theta_{k-1}] + [\delta \theta_k])$ |
| **Backward Propagation** | | |
| 1. $[x_{k-1}]$ | $=$ | $[x_{k-1}] \bigcap ([x_k] - [\delta s_k] \cos([\theta_{k-1}] + \frac{[\delta \theta_k]}{2}))$ |
| 2. $[z_{k-1}]$ | $=$ | $[z_{k-1}] \bigcap ([z_k] - [\delta s_k] \sin([\theta_{k-1}] + \frac{[\delta \theta_k]}{2}))$ |
| 3. $[\delta s_k]$ | $=$ | $[\delta s_k] \bigcap (([x_k] - [x_{k-1}]) / \cos([\theta_{k-1}] + \frac{[\delta \theta_k]}{2}))$ |
| 4. $[\delta s_k]$ | $=$ | $[\delta s_k] \bigcap (([z_k] - [z_{k-1}]) / \sin([\theta_{k-1}] + \frac{[\delta \theta_k]}{2}))$ |
| 5. $[\theta_{k-1}]$ | $=$ | $[\theta_{k-1}] \bigcap ([\theta_k] - [\delta \theta_k])$ |
| 6. $[\theta_{k-1}]$ | $=$ | $[\theta_{k-1}] \bigcap (\arccos(([x_k] - [x_{k-1}]) / [\delta s_k]) - \frac{[\delta \theta_k]}{2})$ |
| 7. $[\theta_{k-1}]$ | $=$ | $[\theta_{k-1}] \bigcap (\arcsin(([z_k] - [z_{k-1}]) / [\delta s_k]) - \frac{[\delta \theta_k]}{2})$ |
| 8. $[\delta \theta_k]$ | $=$ | $[\delta \theta_k] \bigcap (\arccos(([x_k] - [x_{k-1}]) / [\delta s_k]) - [\theta_{k-1}])$ |
| 9. $[\delta \theta_k]$ | $=$ | $[\delta \theta_k] \bigcap (\arcsin(([z_k] - [z_{k-1}]) / [\delta s_k]) - [\theta_{k-1}])$ |
| 10. $[\delta \theta_k]$ | $=$ | $[\delta \theta_k] \bigcap ([\theta_k] - [\theta_{k-1}])$ |

*3.3. Bounded-Error Landmark Parameterization*

A landmark (or feature point) is a 3D point in the global world. The map of the environment thus can be represented by $n$ stationary landmarks: $\mathcal{M} = (\mathcal{L}_1, \mathcal{L}_2, \cdots, \mathcal{L}_n)^T$. Drocourt [24] proposed a parameterization method in the context of interval analysis by representing the landmark with a 3D interval box ($[\mathcal{L}_i] = [x] \times [y] \times [z]$). However, this representation turns out to suffer the envelope problem [25]. To improve the mapping result, we make an extension of the inverse depth parameterization [26] to obtain an bounded-error model to parameterize the landmark. Each landmark

$\mathcal{L}_i$ is defined as a six-dimensional interval vector: $([x_c], [y_c], [z_c], [\alpha_i], [\varphi_i], [d_i])^T$, which models the estimated landmark position at:

$$[\mathcal{L}_i] = ([x_c], [y_c], [z_c])^T + [d_i] \cdot [\mathbf{m}]([\alpha_i], [\varphi_i]), \tag{12}$$

where $[d_i]$ is the unknown depth of the landmark. The coordinate $[x_o], [y_o], [z_o]$ represents the position of the optical center in the world frame when the landmark was seen for the first time; $[\alpha_i]$, $[\varphi_i]$ represent respectively the azimuth and elevation angle for the ray that traces the landmark and $[d_i]$ is the depth of the landmark. $[\mathbf{m}]([\alpha_i], [\varphi_i])$ is a unitary vector pointing from the optical center to the landmark $[\mathcal{L}_i]$, see illustration in Figure 4.

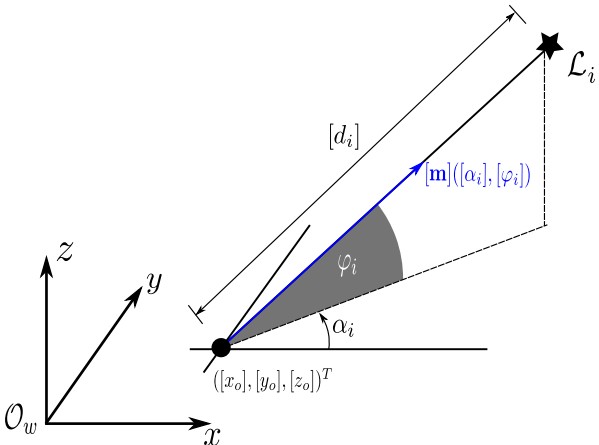

**Figure 4.** Illustration of the ray tracing the landmark.

Since all the parameters are represented by intervals, and $[d_i]$ can be initialized as $[0, +\infty]$, the landmark's uncertainty is modeled as an infinite cone (Figure 5 shows the initialization of two landmarks with different cone uncertainty). It is a realistic representation for the monocular vision uncertainty. The major advantage is that the initialization of $[d_i]$ is undelayed, guaranteed and efficient for landmarks over a wide range as it always includes all possible values without any a priori information.

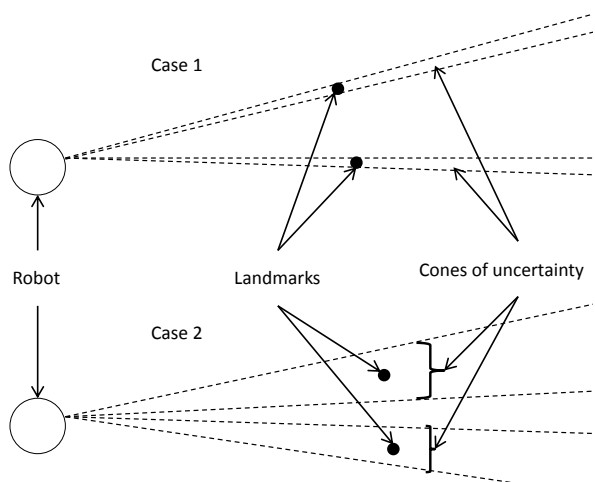

**Figure 5.** Bounded-error landmark initialization.

### 3.4. ICSP Based Estimation

The observation of a landmark is the coordinates of the pixel on the image. Although the depth information is lacking, each of the landmark's position can be estimated through multiple parallax

observations. To achieve the estimation, we form the estimation process as the problem of solving ICSP. Considering the *i*th observed landmark $[\mathcal{L}_i]$, due to its position uncertainty, its projection on the current image plane is no longer a feature point but an area that can be computed via a pin-hole camera model [27]:

$$[\mathcal{L}_i^c] = [R^{wc}]([\mathcal{L}_i] - [\mathcal{X}_k]),\tag{13}$$

$$[S_i] = CamProject([\mathcal{X}_k], [\mathcal{L}_i^c]).\tag{14}$$

The resulting projection $[S_i] = ([u_{\text{pre}}^i], [v_{\text{pre}}^i])^T$ is a bounding area, which is assumed to contain the observed feature point related to the landmark. On this bounding area, we can search for the matching candidates. Probabilistic methods work in the same manner but with an uncertainty ellipsoid. We adopt Dan's method [28], which performs a two-stage matching process (considering both the Euclidean distance and dominant orientation information of feature descriptors) to achieve robust and accuracy matching results. A failure counter and a Zero Normalized Cross Correlation (ZNCC) [29] score are assigned to each landmark to evaluate the stability of the landmark. When the matching candidate is found, a link between the prediction and observation data are built, which can be expressed as an interval domain intersection:

$$\begin{cases} [u_{\text{pre}}^i] = [u_{\text{pre}}^i] \cap [u_{\text{obs}}^i], \\ [v_{\text{pre}}^i] = [v_{\text{pre}}^i] \cap [v_{\text{obs}}^i]. \end{cases}\tag{15}$$

Based on these two top level constraints, we can establish an ICSP that contains a set of nine variables (e.g., $([u_{\text{obs}}^i], [v_{\text{obs}}^i])^T$ , $[\mathcal{X}_k]$ and $[\mathcal{L}_i]$). By solving the ICSP, the robot pose and landmark's position can be contracted. The ICSP is defined as follows:

- A set of nine variables:

$$V_k = \{x_k, y_k, \theta_k, x_c, y_c, z_c, \alpha_i, \varphi_i, d_i\},\tag{16}$$

- A set of nine domains:

$$D_k = \{[x_k], [y_k], [\theta_k], [x_c], [y_c], [z_c], [\alpha_i], [\varphi_i], [d_i]\},\tag{17}$$

- A set of two top level constraints:

$$C_k = (C_k^1, C_k^2)^T,\tag{18}$$

$$\begin{cases} C_k^1 \Leftrightarrow [u_{pre}^i] = [u_{pre}^i] \cap [u_{obs}^i], \\ C_k^2 \Leftrightarrow [v_{pre}^i] = [v_{pre}^i] \cap [v_{obs}^i]. \end{cases}\tag{19}$$

Such ICSP can be generated for each matched landmark, and can be independently solved by a local consistency ICP algorithm. After multiple observations from different parallaxes at different times, the ICSP building and solving process will be repeatedly performed. The position of each landmark thus can be estimated.

## 3.5. Dynamic ICSP Graph (DIG) Optimization

The notion of graph optimization has been widely used in a Simultaneous Localization and Mapping (SLAM) area. In this section, we will present our proposed graph optimization method in the context of interval analysis.

### 3.5.1. Graph Architecture

As we presented in our previous study [30], the generated ICSPs can be solved independently by local consistency ICP algorithms (Forward/Backward, HC3, HC4, etc.). However, we note that, at each timestep *k*, the generated ICSPs associated with each observed landmark are linked together by two links. The first one is the spatial link; at each timestep, the generated ICSPs are linked together

by the variables $x_k$, $y_k$, $\theta_k$ (the current robot pose state). Another one is the time link between the robot pose state at different timesteps. Based on these two links, an ICSP graph can be built. In order to be consistent with the traditional graph notion, which consists of vertices and edges, in our proposed ICSP graph, the vertex represents the robot pose or landmark position while the edge represents the constraints of the ICSP imposed on the connected vertices. Figure 6 gives a simplified illustration of the ICSP graph. Denote the ICSP graph by $G_k(V, E)$; each edge is given by

$$E_k^{n_x} \leftarrow \mathrm{ICSP}_k^{n_x}(V_k, D_k, C_k), \tag{20}$$

$$E_{k-1}^{k} \leftarrow \mathrm{ICSP}_{k-1}^{k}(V_k, D_k, C_k), \tag{21}$$

where $n_x$ is the index of landmark and $k$ is the timestep. $\mathrm{ICSP}_k^{n_x}$ denotes the generated ICSP corresponding to the observation of landmark $n_x$ at timestep $k$, while $\mathrm{ICSP}_{k-1}^{k}$ denotes the ICSP generated in the motion estimation process. By constructing the ICSP graph, the domains of all the variable are connected together. Our idea is that an improvement of one vertex's domain estimation due to new observations can be propagated to contract the other vertex's domain via the graph architecture, and this could be a bi-directional process since ICP algorithms always work in a forward and backward propagation manner.

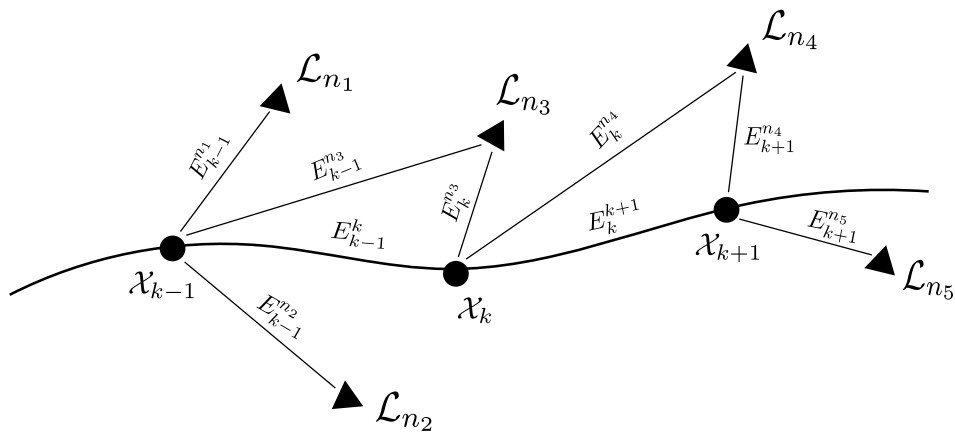

**Figure 6.** ICSP graph structure.

To the best of our knowledge, this is the first time that the notion of an ICSP graph is proposed to solve the robotic localization and mapping problem. The ICSP graph can be used for global and intensive domain contraction, as it brings together all of the measurement information. At each timestep, when new observation has been processed, the corresponding ICSPs are added to the graph and old ones will be removed.

### 3.5.2. Graph Operation: Augmentation and Depletion

The augmentation and deletion actions of the ICSP graph actually work on the variables and constraints. The union of an ICSP $P = \{V, D, C\}$ and a constraint $K$ (which only contains variables that are already presented in $P$) is defined by:

$$P' = P \cup K = \{V, D, C \cup K\}. \tag{22}$$

Similarly, the union of an ICSP $P_1 = \{V_1, D_1, C_1\}$ and another ICSP $P_2 = \{V_2, D_2, C_2\}$ is achieved by union the variable, domain and constraint sets:

$$P' = P_1 \cup P_2 = \{V_1 \cup V_2, D_1 \cup D_2, C_1 \cup C_2\}. \tag{23}$$

The union operation of ICSPs can be directly used to make augmentation to the ICSP graph. The depletion of an ICSP from the ICSP graph is a bit different. Denote the ICSP network by $\mathbb{P} = \{\mathbb{V}, \mathbb{D}, \mathbb{C}\}$; to remove the ICSP $P_1 = \{V_1, D_1, C_1\}$ from $\mathbb{P}$, one should delete the constraint set $C_1$ and the variables $V_1'$ that only appear in $P_1$ ($V_1' \in V_1$ and $V_1' \cap \{\mathbb{V}/V_1\} = \varnothing$). $V_1'$ can be calculated by

$$V_1' = V_1/\{\mathbb{V} \cap V_1\}. \tag{24}$$

Then, the new ICSP graph $\mathbb{P}'$ is given by

$$\mathbb{P}' = \mathbb{P}/P_1 = \{\mathbb{V}/V_1', \mathbb{D}/D_1', \mathbb{C}/C_1\}. \tag{25}$$

### 3.5.3. Dynamic Graph Optimization

The entire ICSP graph can be regarded as a stack of ICSPs which can be solved by Interval Constraint Propagation Algorithms (in our case, we use the HC4 algorithm). However, solving the entire graph at each timestep is a complex and time consuming task. To maintain the real time adaptability, we propose to solve the ICSP graph in a sliding window $\Xi$: at each timestep $k$; only the sub-graph $G_{k-\Xi \to k}(V, E)$ is taken into computation. When $\Xi = 1$, it is equivalent to solve the multiple ICSP:

$$\nu\text{ICSP}_k \leftarrow \sum_{n_x \in N_{k-1}} \text{ICSP}_{k-1}^{n_x} + \sum_{n_x \in N_k} \text{ICSP}_k^{n_x} + \text{ICSP}_{k-1}^k, \tag{26}$$

where $N_{k-1}$ and $N_k$ are respectively the index set of the observed landmarks at timestep $k-1$ and $k$. By solving the $\nu\text{ICSP}_k$, a better estimation of the landmark's parameters could be achieved. This can be viewed as a graph optimization process, which we named the Dynamic ICSP Graph Optimization (DIGO).

## 4. Online Localization

### 4.1. Real-Time Localization

After the visual teach phase, a map consisting of a set of landmarks is obtained. Then, in the real-time localization phase, the robot could locate itself by taking advantage of the map. This is done by matching the detected feature point with the landmarks in the map and finding out the 3D–2D correspondences. Each correspondence imposes constraints on the robot pose domain and the landmark domain, from which we can set up contractors (refer to Sections 2.4 and 3.4) to contract the robot pose domain. For the sake of simplicity, assume that $[\mathcal{X}_k]$ is the estimated robot pose domain obtained from the motion estimation process and three landmarks ($\mathcal{L}_{n_1}, \mathcal{L}_{n_2}, \mathcal{L}_{n_3}$) are matched to the detected feature points, see Figure 7, then we can build three contractors to contract the robot pose domain to an optimal set $\mathbb{X}_k$:

$$\mathbb{X}_k = \mathcal{C}_{n_1}([\mathcal{X}_k]) \cap \mathcal{C}_{n_2}([\mathcal{X}_k]) \cap \mathcal{C}_{n_3}([\mathcal{X}_k]). \tag{27}$$

The basic idea is to contract the domains of the vehicle pose by using all of the available contractors successively. According to the algebraic operation rules of contractors defined in [10], Equation (27) can be transformed as:

$$\mathbb{X}_k = (\mathcal{C}_{n_1} \cap \mathcal{C}_{n_2} \cap \mathcal{C}_{n_3})([\mathcal{X}_k]). \tag{28}$$

It works like a new contractor that inherits the qualities of its constituents. The real-time localization result can be calculated by an interval hull operation that generates a minimal interval box to include all the values of the optimal set $\mathbb{X}_k$. Such interval box is regarded as the final robot pose domain:

$$[\mathcal{X}_k^*] \leftarrow Hull(\mathbb{X}_k). \tag{29}$$

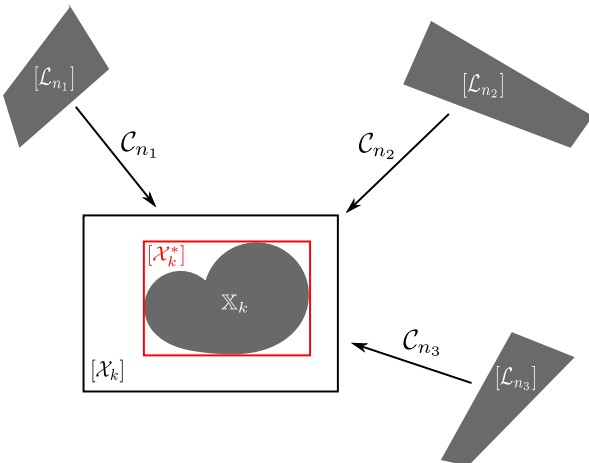

**Figure 7.** Contraction of a robot pose domain.

### 4.2. Post-Localization

The real-time localization result is obtained by using only the current knowledge of the observations; this localization result can be further improved by considering a dynamic ICSP graph (see Section 3.5) in a sliding window during each timestep, such that the localization improvement due to the new observation can be propagated to the past through the graph architecture and improve the past localization. As a result, an improved localization result could be obtained, which we call 'post localization'.

### 4.3. Consistency Evaluation

An estimation result is said to be consistent if it includes the real value. In order to evaluate the consistency of the localization results that appear to be intervals, interval error is defined. It is calculated by the upper and lower bound of the estimated state minus the reference value:

$$[\psi_k(\alpha)] = [\underline{\alpha}_k - \alpha_k^*, \overline{\alpha}_k - \alpha_k^*], \tag{30}$$

where the interval $[\alpha_k] = [\underline{\alpha}_k \overline{\alpha}_k]$ is the estimated state of variable $\alpha$ ($\alpha \in \{x, y, z\}$) at timestep $k$, and $\alpha_k^*$ is the reference value of parameter $\alpha$. The width of interval error $w([\psi_k(\alpha)])$ can be used to evaluate the estimation accuracy. The consistency can be determined by testing whether the resulting interval error contains the zero value. Mathematically:

$$0 \in [\psi_k(\alpha)] \iff \alpha_k^* \in [\alpha_k]. \tag{31}$$

If $\alpha$ is a vector variable, then the criterion is applied to verify the consistency of every components of $\alpha$. If all the components reach consistency, then the estimation of $\alpha$ is said to be consistent. Moreover, if $\forall k \in \{1, 2, \cdots, \infty\}$, $\alpha_k^* \in [\alpha_k]$ is satisfied, then the estimation method is said to be a consistent one as it could provide consistent results all the time.

## 5. Results and Discussion

### 5.1. Numerical Results

To test the feasibility of the proposed ICSP graph based visual teach method, we set up a simple numerical experiment. Assume that the robots start from the origin and move straight forward in the positive direction of the *x*-axis with constant linear and angular velocity ($v = 0.1 \text{m} \cdot \text{s}^{-1}$, $\omega = 0$). 5 landmarks are positioned in the surrounding environment without any prior knowledge of their positions. At each timestep, the robot detects the five landmarks with a simulated camera model

($[fk_u, fk_v, c_u, c_v] = [320, 320, 320, 240]$, image size: $640 \times 480$, pixel error: 1 pixel) and focus on estimating their positions.

Figure 8 displays, since initialization, the evolution of interval cone uncertainty of the four landmarks in top view. At timestep $t = 1$, the four landmarks are firstly detected by the robot and the initialization procedure is conducted. The domains of each landmark are initialized as an infinite cone due to the lack of information of the landmark depth. With the robot moving forward, parallax observations to each landmark are generated. For each observation, a new ICSP is created and solved via the HC4 algorithm. By dynamically constructing and solving ICSP graphs, each of the landmarks' domains are contracted and converged to their true positions when enough parallax is eventually available. To evaluate the final mapping result, it is required to convert the interval cone representation to a global 3D position box using Equation (12), such that the uncertainty of each landmark can be calculated. Here, we use the mean position-box volume (MPV) of the four landmarks to indicate the accuracy the map, which is calculated by:

$$\text{MPV} = \frac{1}{n} \sum_{i=1}^{n} (w([x_i]) * w([y_i]) * w([z_i])), \tag{32}$$

where $n$ is the number of the converged landmarks (in the simulation case, $n = 4$). $(x_i, y_i, z_i)$ is the 3D position of the $i^{th}$ landmark. MPV can be used as an indicator to evaluate the precision of the obtained map.

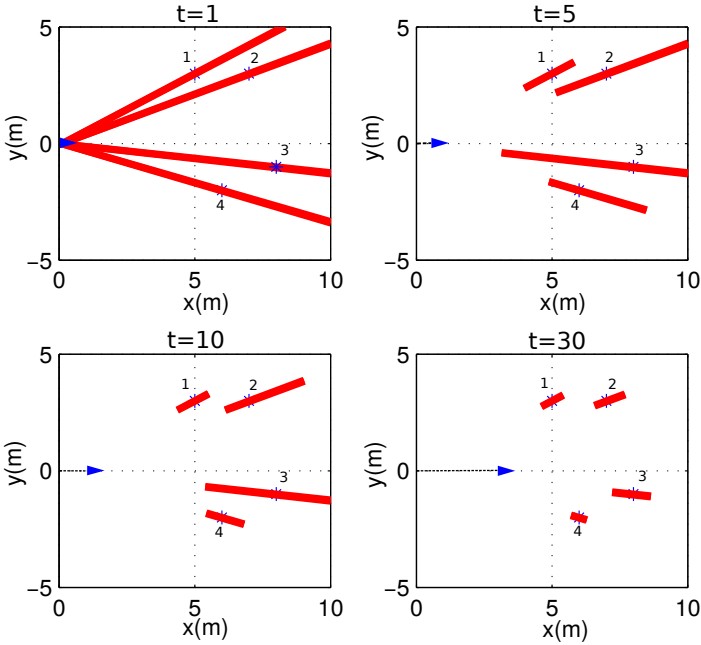

**Figure 8.** Evolution of a landmark's interval cone uncertainty.

Figure 9a shows how the window size influences the mapping accuracy of our proposed DIGO method. As expected, the bigger the window size of the ICSP graph, the better mapping result could be obtained. The price to pay is the computation burden. Figure 9b shows that the average processing time of each timestep increases linearly with the increment of window size. In practical application, a compromise between the mapping precision and processing time should be made.

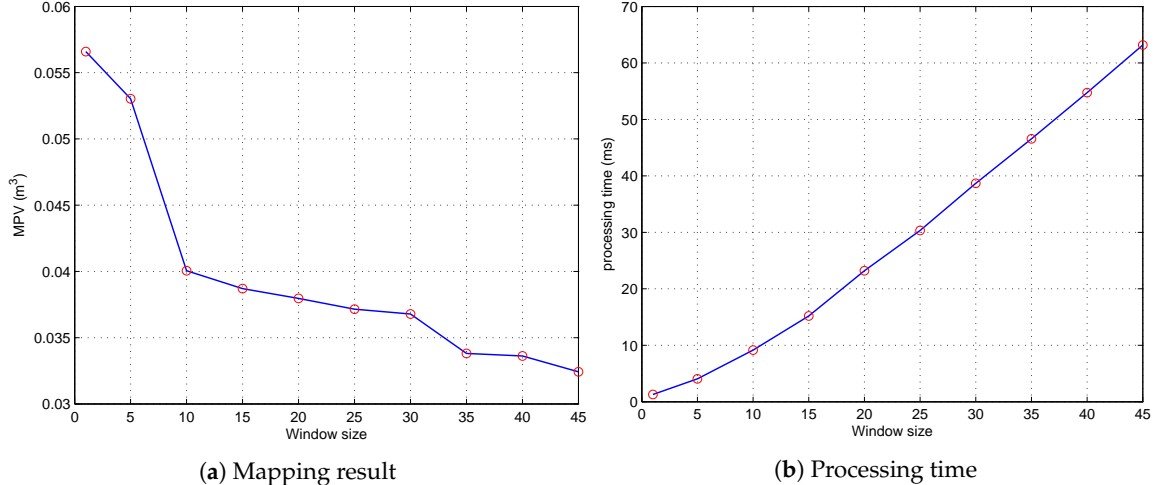

(**a**) Mapping result  (**b**) Processing time

**Figure 9.** Mapping result and processing time under different window size.

To manifest the improvement of our proposed DIGO method with the traditional one [30] that solves the ICSPs locally (Local ICP Contraction: LIC), we make a comparison of the final estimated landmark's position. For the sake of simplicity, the two methods perform independently with the same data source and the final mapping results are compared by calculating the improvement factor $\eta$ in terms of axis $x$, $y$ and $z$:

$$\eta(\alpha) = \frac{\mid \text{LIC}(\alpha) - \text{DIGO}(\alpha) \mid}{\text{LIC}(\alpha)} \times 100\%, \tag{33}$$

where $\alpha \in \{x, y, z\}$ represents the axis estimation. For this comparison, the window size of DIGO is set to be 10. The results are detailed in Table 2, and the mean dimension widths (MDW) of all all landmarks on each axis are calculated. As we can see, our proposed DIGO method expresses better estimation results than the LIC method. This is because the DIGO method could take advantage of measurements from other landmarks to benefit the current landmark. According to our previous study [30], the bigger the parallax is, the better the landmark's estimation will be. During each observation, different landmarks may have different parallax. Some of them (bigger parallax) could obtain very good estimations while others (smaller parallax) are not. By using the ICP algorithm, the contraction of one landmark's parameter domain can be propagated to improve the others thanks to the ICSP graph architecture.

**Table 2.** Comparison of mapping results.

| MDW | $x$ | $y$ | $z$ |
|---|---|---|---|
| LIC (*m*) | 0.117 | 0.068 | 0.024 |
| DIGO (*m*) | 0.103 | 0.061 | 0.022 |
| $\eta$ | 12% | 10.3% | 8.3% |

*5.2. Real Data Set Experiment*

To evaluate the performance of our method, we used the open source experimental data set from Institute Pascal. Data were collected on the VIPALAB platform (see Figure 10a), driving on the PAVIN track (see Figure 10b), an experimental site located on the campus of Blaise Pascal University. The platform was equipped with multiple sensors, providing multisensory timestamped data that can be used in a large variety of applications. In our experimentation, we use the set of timestamped data from odometers, gyro and the front mounted monocular camera. The real-time kinematic GPS (RTK-GPS) data (accuracy $\pm 1$ cm), which provide absolute localization measurement, are regarded as the ground truth and are used in the mapping phase. Readers could refer to [31] to know more details

about the data set (Data used in this manuscript are openly available at "http://ipds.univ-bpclermont.fr/").

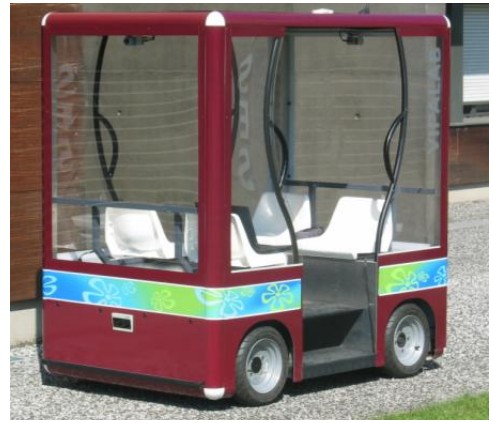

(**a**) VIPALAB platform

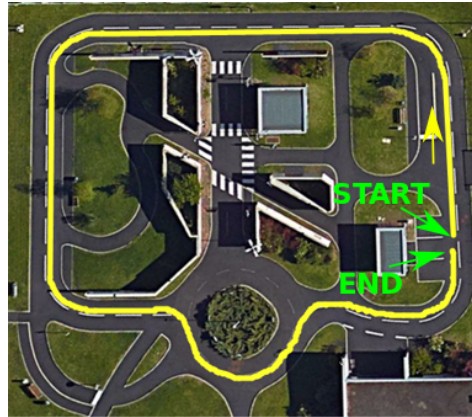

(**b**) The PAVIN track

**Figure 10.** Institute Pascal Experimental Environment.

A mapping stage is firstly conducted. The car-like robot navigates around the environment and establishes a map of its surroundings with the help of RTK-GPS measurements. Every step in the mapping and localization process relies on feature detection and matching. We use the Speed Up Robust Features (SURF) algorithm [32], recommended by Schmidt [33] after a comparison between several feature detection algorithms in the context of the SLAM algorithm, showing that SURF provides results with a very low error rate. Each detected feature point is initialized with proposed bounded-error parameterization, and the observation error is set to be one pixel. Figure 11 describes the distribution of the position box volume (uncertainty) of each landmark in the built map and the statistical results are shown in Table 3. It can be seen that most of the landmarks (97%) obtain very good estimation, uncertainty less than 0.008 m$^3$, demonstrating the ability of our method to obtain a converged map. The actual result of our proposed method should be better than this since the interval position box is a pessimistic representation of the interval cone uncertainty (envelope problem).

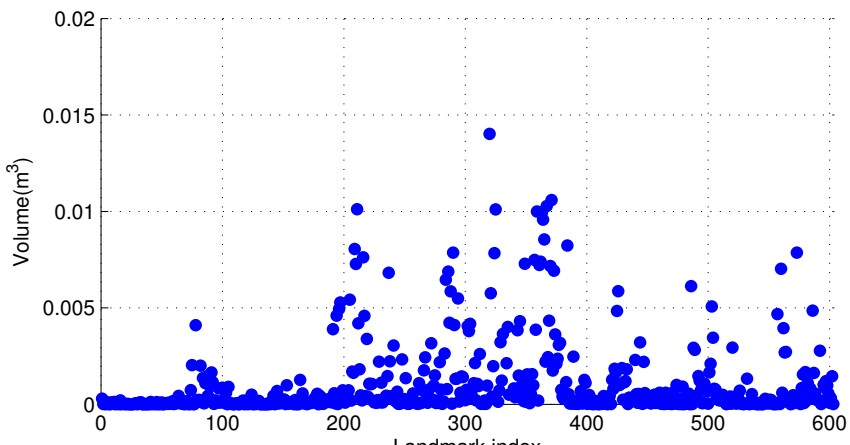

**Figure 11.** Landmark uncertainty statistics.

**Table 3.** Statistics of landmark location uncertainty.

| Vol (m$^3$) | <0.001 | <0.004 | <0.008 |
|---|---|---|---|
| Percentage | 75% | 91% | 97% |

After the map is established, the map data are then used for online localization. In this stage, only the monocular camera is used for sensing the environment. The RTK-GPS data are treated as the reference data for comparison. On the real-time localization stage, by dynamically detecting and matching detected landmarks with the map data, which offers constraints on the robot position, the robot follows a nearly 200 m trajectory. To evaluate the consistency of the localization result, the real-time interval error is calculated. At each timestep $k$, when the new captured image is processed and localization procedure is done, $\psi_k(x)$ and $\psi_k(y)$ are calculated, respectively. Figure 12 depicts the interval corridors consisting of the upper and lower bounds of $\psi_k$. It is clear that the zero lines are well included by those corridors, proving that the localization results are consistent all along the track. The localization boxes output by our method is trustworthy; it encloses the real robot position all over the time. This is a significant result in robotic localization applications where safety is a crucial issue. Computing an average interval width of the corridors gives 11.7 cm accuracy for $x$ and 13.4 cm for $y$. Closer framework has already been proposed by different researchers [17,18]. Eric Royer [17] uses bundle adjustment in the mapping step and the localization results are obtained via a Newton iteration method. Their method gives an average localization error of 15 cm. Courbon's method [18] achieves an average lateral error of 25 cm on an urban vehicle navigating along a 750 m trajectory. Our method could obtain a compatible localization accuracy. The strength of our method is that the output localization boxes are guaranteed to include the robot's real position.

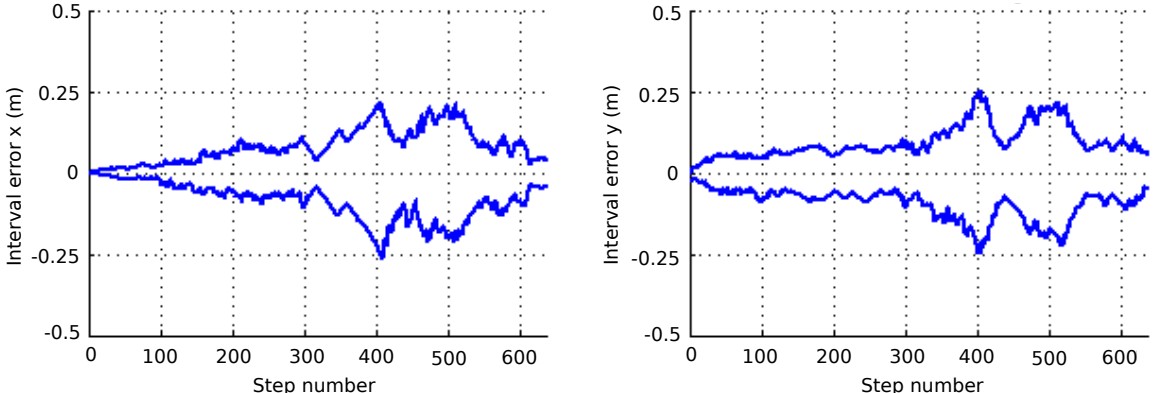

**Figure 12.** Interval error of real time localization.

Figure 13 shows the post localization (PL) result under the window size $\Xi = 50$, which means that, at each localization step, the ICP algorithm will perform backward propagation through the ICSP graph architecture for 50 steps. As we can see from the figure, the resulting localization still remains consistent while the width of the interval error corridor has been reduced. It demonstrates that the real-time localization (RTL) result can be further improved by our proposed DIGO method. This is reasonable; as we state in Section 3.5.3, the estimation of current pose estimation can be propagated backward to incorporate with the past observations. The ICSP graph can benefit from this incorporation and obtain an optimized estimation of the pose state. Since the ICP algorithm is a type of bi-directional method, this improvement in the past is then propagated to the current timestep and helps to improve the real-time localization result.

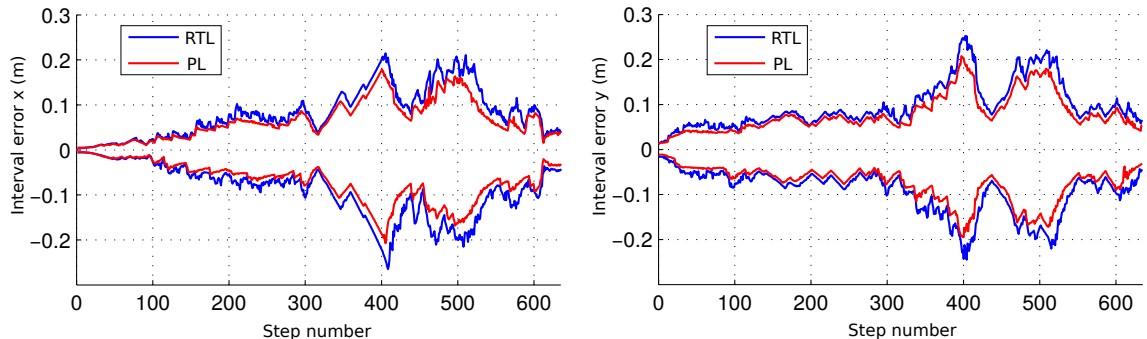

**Figure 13.** Comparision of real-time localization and post localization.

## 6. Conclusions

To deal with the inconsistency problem in robotic localization and mapping area, this paper proposed an ICSP graph based localization framework that includes a visual teach and repeat stage. By dynamically constructing and solving the ICSP graph, a consistent localization result is obtained. Both simulation and real data set experimentation results have been presented, showing the feasibility of interval analysis in solving the mapping problem with GPS-support and the use of an ICSP graph can enhance the landmark's estimation process and leads to a better map. Recall the two-stage process: visual teach and repeat; we address here that the map building process is a little resource consuming (RTK-GPS sensors are needed) but is not meaningless since the built map can be beneficial for other robots that intend to achieve reliable localization in the environment. Another advantage of our method is that the output localization boxes are guaranteed to enclose the real position of the robot. This reliability property can support high level applications in decision-making where safety is the most crucial issue. Future work will focus on the visual teach stage in terms of precision and consistency.

**Author Contributions:** Conceptualization, Z.W. and A.L.; Data curation, Z.W.; Formal analysis, Z.W.; Funding acquisition, X.Z.; Investigation, Z.W.; Methodology, Z.W.; Project administration, A.L.; Resources, A.L.; Software, Z.W.; Supervision, A.L.; Validation, Z.W.; Visualization, Z.W.; Writing— original draft, Z.W.; Writing—review & editing, Z.W., A.L. and X.Z.

**Funding:** This research received no external funding.

**Conflicts of Interest:** The authors declare no conflict of interest.

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
