# Peer review of "Dynamic ICSP Graph Optimization Approach for Car-Like Robot Localization in Outdoor Environments"

_computers, doi:10.3390/computers8030063_

Round 1

Reviewer 1 Report

Nice work, though many explanations of equations could be more clearer or perhaps expanded more.

Idea as a whole is interesting to me, details maybe not so much, relevant
from the point that so many research institutions are working on these same problems though using often very differing methods.

The topic is not so highly original, as there is so much work being made on the topic.... there are several competing methodologies for this... and this is
more of an improvement then novelty.

compared with other published material:
In details less... at least for me it adds more as a concept. Though I
could not detect plagiarism... the publishing rate is so fast in this field
that what they add to other material is a difficult question.... I think
not so much in the details.

structure:
Text is very clear, though equations and their explanations are
difficult to follow sometimes. And the text clear and easy to read.

Based on the experimental results, the conclusions consistent with the evidence and arguments presented.

However, there does not seem to be enough clear comparisons of results to other methodologies.

Author Response

Nice work, though many explanations of equations could be more clearer or perhaps expanded more.

Idea as a whole is interesting to me, details maybe not so much, relevant
from the point that so many research institutions are working on these same problems though using often very differing methods.

The topic is not so highly original, as there is so much work being made on the topic.... there are several competing methodologies for this... and this is
more of an improvement then novelty.

compared with other published material:
In details less... at least for me it adds more as a concept. Though I
could not detect plagiarism... the publishing rate is so fast in this field
that what they add to other material is a difficult question.... I think
not so much in the details.

--Thanks so much for your comments on the manuscript. Indeed, the topic of this paper is not so highly original, there are a lot of works focus on this topic and a lot of publications have been issued. The main contribution of this paper is that it propose a deterministic method (using interval analysis theory) which is quite different from most of the existing methods to deal with the well know inconsistency problem. In addition, it presented an ICSP graph architecture in the context of graph theory but using interval constraint propagation algorithm to conduct the optimization process.

structure:
Text is very clear, though equations and their explanations are
difficult to follow sometimes. And the text clear and easy to read.

-- I am sorry about this. As we know, interval analysis is an alternative tool for numerical calculation and not so widely studied by the community. Due to the length limitations, I only presented some fundamental introductions to the theory basis, and I would like to make this paper focus more on the methodology and technical part. So that's why some equations are not so detailedly explained.

Based on the experimental results, the conclusions consistent with the evidence and arguments presented.
However, there does not seem to be enough clear comparisons of results to other methodologies.

-- Thanks for you comments. I add a paragraph to discuss the performance of proposed method with other published methods and make a comparison with them. The following paragraph is added to the manuscript:

"Closer framework have already been proposed by different researchers [1], [2]. Eric Royer et al. [1] uses bundle adjustment in mapping step and the localization results are obtained via Newton iteration method. Their method gives an average localization error of 15 cm. J. Courbon's method [2] achieves an average lateral error of 25 cm on an urban vehicle navigating along a 750 m trajectory. Our method could obtained a compatible localization accuracy. The strength of our method is that the output localization boxes are guaranteed to include the robot's real position. "

[Royer2005] Eric Royer, Maxime Lhuillier, Michel Dhome, and Thierry Chateau. Localization in urban environments: monocular vision compared to a differential gps sensor. In IEEE Computer Society Conference on Computer Vision and Pattern Recognition, volume 2, pages 114–121, 2005.

[Courbon2009] Jonathan Courbon, Youcef Mezouar, and Philippe Martinet. Au- tonomous navigation of vehicles from a visual memory using a generic camera model. IEEE Transactions on Intelligent Transportation Systems, 10(3):392–402, 2009.

Reviewer 2 Report

The authors propose a new method to deal with the inconsistency problem in robotic localization and mapping area. They introduce an ICSP graph based localization framework which includes visual teach and repeat stage. By dynamically constructing and solving the ICSP graph, a consistent localization result is obtained. Both simulation and real data set experimentation results have been presented, demonstrating the feasibility of their approach. The are just some typos and issues in the manuscript to be addressed (line number is indicated):

11: show -> showing

20: which based -> which is based

24: method -> methods

32: produced -> produce, error -> error values

38: as -> as is

42: extra space after "receiver"

block 171: in no long -> is no longer, assumed to contains -> assumed to contain

block 183: at different timestep -> at different timesteps, vertices and edges (comma instead of full stop)

block 201: Interval -> interval

204: an -> a

block 207: map consists of -> map consisting of

209: an -> a

237: sovled -> solved

248: could -> could be

278: landmark -> landmarks

285: landmark -> landmarks

289: consist -> consisting

301: an -> a

302: current -> current time step

313: lanmark -> landmarks

The acronym MDW in table 2 is not introduced in the text. Same for RTK-GPS in line 267.

Reference number 25 point to a non-english manuscript, it is better in my opinion to reference an english source or to briefly recall the concept (the envelope problem) in this manuscript.

Author Response

The authors propose a new method to deal with the inconsistency problem in robotic localization and mapping area. They introduce an ICSP graph based localization framework which includes visual teach and repeat stage. By dynamically constructing and solving the ICSP graph, a consistent localization result is obtained. Both simulation and real data set experimentation results have been presented, demonstrating the feasibility of their approach. The are just some typos and issues in the manuscript to be addressed (line number is indicated):

-- Thanks so much for your comments on the manuscript. I have already corrected all the typos. 

Reference number 25 point to a non-english manuscript, it is better in my opinion to reference an english source or to briefly recall the concept (the envelope problem) in this manuscript.

-- Thanks for your advise. I do agree with you. It is better to refer to an english publication. I add a new reference which  recall the envelope problem. 

[Keafott1996] Kearfott, R.B. Rigorous global search continuous problems; Springer, 1996.